# Potential Protectivity of a Conjugated COVID-19 Vaccine against Tetanus

**DOI:** 10.3390/vaccines12030243

**Published:** 2024-02-26

**Authors:** Delaram Doroud, Fatemeh Ashrafian, Amir Javadi, Sarah Dahmardeh, Mohammad Banifazl, Anahita Bavand, Mona Sadat Larijani, Amitis Ramezani

**Affiliations:** 1Quality Control Department, Production and Research Complex, Pasteur Institute of Iran, Tehran 3159915111, Iran; d_doroud@pasteur.ac.ir; 2Clinical Research Department, Pasteur Institute of Iran, Tehran 1316943551, Iran; f_ashrafian@pasteur.ac.ir (F.A.); a_bavand@pasteur.ac.ir (A.B.); 3Department of Community Medicine, School of Medicine, Qazvin University of Medical Sciences, Qazvin 1985717413, Iran; javadi_a@yahoo.com; 4Vaccination Department, Pasteur Institute of Iran, Tehran 1316943551, Iran; s_dahmardeh@pasteur.ac.ir; 5Iranian Society for Support of Patients with Infectious Disease, Tehran 1989934148, Iran; iiccom@iiccom.org

**Keywords:** anti-tetanus immunity, COVID-19, vaccine, carrier protein, tetanus toxoid

## Abstract

PastoCovac is a subunit protein vaccine against COVID-19 which contains the tetanus toxoid as a carrier conjugated to SARS-CoV-2 RBD. The primary goal of the tetanus application was to elicit a stronger specific response in the individuals. However, conjugate vaccines have the potency to generate anticarrier antibodies in addition to the target antigen. Therefore, the present study aimed to evaluate the PastoCovac vaccine in the humoral immune induction against tetanus. Six groups of individuals, including those who received one, two, or three doses of the PastoCovac vaccine, Td vaccine, and also the controls who received other COVID-19 vaccines (except PastoCovac), were investigated. The anti-tetanus IgG was assessed by an ELISA assay in all vaccinated groups. The antibody persistency against tetanus in the group who received one dose of the PastoCovac vaccine was also assessed on day 60, 90, and 180 after the last injection. The anti-tetanus antibody titer in the three groups of PastoCovac recipients was positive, though additional doses of the vaccine led to a significant antibody rise (*p* = 0.003). Notably, the comparison of the mean antibody titer between the Td recipients and those who received one/two doses of PastoCovac showed that the mean rise in the antibody titer before and after the injection was not significant. Although the antibody titer on day 180 decreased to a lower level than on day 21, it was still estimated to be highly positive against tetanus. Eventually, none of the PastoCovac recipients presented vaccine side-effects during the follow-up. The current data indicate that the tetanus conjugate vaccine against COVID-19, PastoCovac, could induce immune responses against tetanus, which can persist for at least 6 months. Combination vaccine formulae containing TT and DT as carriers for conjugate vaccines could be considered instead of TT and/or DT boosters in adults if they are indicated.

## 1. Introduction

SARS-CoV-2 caused the most recent pandemic, COVID-19, which led to the rapid design of vaccines and clinical trials in order to inhibit the global spread of the virus worldwide. Generally, vaccines could be categorized into five main platforms, including live-attenuated or inactivated viruses, nucleic acids (RNA or DNA), recombinant protein, viral vectors, and virus-like particles (VLPs) [1,2]. Vaccines are ideally supposed to induce immune responses through both humoral and cellular arms. Therefore, different approaches, including adjuvants, nanoparticles, targeting antigen-presenting cells, heterologous prime–boost regimens, and antigen carriers, have been explored to date against different pathogens [3,4].

Carrier proteins are applied to enhance the immunogenicity of the formulated antigens in vaccine formulae. The approved carrier proteins for human use include tetanus toxoid (TT), diphtheria toxoid (DT), Haemophilus protein D (PD), CRM197 (a nontoxic mutant of diphtheria toxin), and the outer membrane protein complex of serogroup B meningococcus (OMPC) [5,6,7]. TT, as a strong carrier protein, is composed of several T-cell peptides. These universal T-cell epitopes could possibly increase the effectiveness of the carrier [8].

PastoCovac was manufactured and administered in Iran after a successful technology transfer from Cuba, and showed significant efficiency in vaccine studies [9]. This vaccine is a protein subunit one, composed of conjugated RBD to the tetanus toxoid. The conjugation of RBD to TT leads to the effective induction of both humoral and cellular immune responses, owing to the fact that multiple receptor-binding motifs (RBMs) expose where the neutralizing epitopes predominate [10,11]. PastoCovac was primarily administered against COVID-19 as the main-dose vaccine. PastoCovac has been used as the primary series, and also as the booster dose, after inactivated vaccines and adenovirus-based vaccines, since its safety profile was proven in the clinical trial in Iran [9].

It is clear that the carrier proteins as part of the vaccine are immunogenic and elicit a specific anti-carrier response which is measurable by in vitro assays [12]. Here, we aimed to assess the ability of the PastoCovac vaccine to induce antibodies against tetanus in immunized individuals. Moreover, the durability of the humoral immune responses against tetanus was investigated 6 months after the last dose of the PastoCovac vaccine, as well as the vaccine safety.

## 2. Material and Methods

### 2.1. Study Population

This study was conducted at the Pasteur Institute of Iran. All the participants were provided with the written consent form prior to participation. The study protocol was performed according to the Declaration of Helsinki (Fortaleza, 13 October 2013) and was approved by the Ethics Committee of the Pasteur Institute of Iran (ethics code number: IR.PII.REC.1402.020).

The participants were divided into 6 case groups (Figure 1) based on the number of PastoCovac (PCovac)/tetanus vaccine doses.

(1)PCovac-1D: COVID-19-vaccinated individuals who received one dose of the PastoCovac vaccine.(2)PCovac-2D: COVID-19-vaccinated individuals who received two doses of the PastoCovac vaccine.(3)PCovac-3D: COVID-19-vaccinated individuals who received three doses of the PastoCovac vaccine.(4)PCovac + Td (CT): COVID-19-vaccinated individuals who received one dose of the PastoCovac vaccine, followed by a dose of the Td vaccine.(5)Td + PCovac (TC): COVID-19-vaccinated individuals who received a dose of the Td vaccine, followed by one dose of the PastoCovac vaccine.(6)Tetanus: COVID-19-vaccinated individuals from different platforms, except the PastoCovac vaccine, who received a dose of the Td vaccine.

Control: COVID-19-vaccinated individuals from different platforms, except PastoCovac vaccine, who had a prior injection of the Td vaccine > 10 years.

### 2.2. Vaccines

An amount of 0.5 mL of the Td vaccine [2 Lf diphtheria toxoid, 8.8 Lf tetanus toxoid, 1.5 mg aluminum phosphate adsorbed, reduced antigen(s) content, reduced thimerosal, Biological E. Limited, India] was injected intramuscularly in the deltoid.

PastoCovac (Soberana) (25 µg RBD conjugated to 20 µg tetanus toxoid, 0.5 mL aluminium hydroxide) was administered intramuscularly in the deltoid [11].

### 2.3. Immunogenicity Assessment

Antitetanus IgG was assessed by an ELISA assay in all vaccinated groups (Tetanus Elisa IgG kit Vircell G1008, Granada, Spain) and an antibody index > 11 was considered positive against tetanus (OD > 1.2). The antibody persistency against tetanus in the group who received one dose of the PastoCovac vaccine was also assessed on day 60, 90, and 180 after the last injection.

### 2.4. Vaccine Safety

All the case participants were monitored after vaccination by Td or PastoCovac 30 min post-injection. Furthermore, any kind of adverse event was recorded in the questionnaire through the interviews via phone call.

### 2.5. Statistical Analysis

Shapiro–Wilk’s W test was utilized to assess the normality of the numerical data. Descriptive statistics (mean and standard deviation (±SD), median and interquartile range (±IQR), minimum and maximum, or frequency and percentages) were used to represent the data. The one-way ANOVA, *t*-test, and Mann–Whitney U-test were used to compare the means between the groups, where appropriate. The repeated-measures ANOVA was used to assess the mean of the anti-tetanus Ab changes over time. Statistical analyses were performed with GraphPad Prism 8.0 for Windows (GraphPad Software, San Diego, CA, USA) and SPSS (v 18.0; SPSS Inc., Chicago, IL, USA) software.

## 3. Results

The present study included 106 subjects (60 females, 46 males) with the mean age of 43.2 ± 13.4 years old. The number of participants in each defined group was as follows:

PCovac-1D: n = 10; PCovac-2D: n = 15; PCovac-3D: n = 19; CT: n = 10; TC: n = 11; Tetanus group: n = 15; 21 individuals as controls. There was no significant difference between the groups regarding demographic characteristics. An anti-tetanus antibody index > 11 (OD > 1.2) was considered as positive according to the semi-quantitative ELISA test.

Three groups of the individuals who received one, two, or three doses of the PastoCovac vaccine were compared. The mean anti-tetanus antibody rise was significantly different between these groups (*p* = 0.004). The additional statistical test showed that this difference between the PCovac-1D and PCovac-3D groups was significant (*p* = 0.003). The mean rise in the antibody titer in the PCovac-3D group was 36.8 units more than the PCovac-1D group. In other words, the average increase in the antibody titer in the PCovac-3D group was twice that of the PCovac-1D group. The assessment of the anti-tetanus antibody in each group is presented in Table 1 and Table 2.

The comparison of the mean antibody titer between the Tetanus and Pcovac-1D groups showed that the mean rise in the antibody titer during the first (*p* = 0.326) and second (*p* = 0.728) measurements was not significant (before and after the injection, respectively).

Furthermore, comparing the mean antibody titer between the Tetanus and Pcovac-2D groups showed that the difference between the mean titer rise of the two groups was not statistically significant (*p* = 0.128). Nevertheless, the comparison of the Tetanus group and PCovac-3D cases showed that the mean of the antibody titer between two groups was significantly different (*p* < 0.001). On average, the antibody titer rise in the PCovac-3D group was about 35 units more than the Tetanus group. That is to say, the mean titer rise in the PCovac-3D group was about 1.67 times that of the Tetanus group.

The comparison of the antibody mean titer between the Tetanus and CT groups showed that the antibody rise in the CT group was about 1.3 times higher than the Tetanus group, although the difference was not statistically significant (*p* = 0.1).

Comparing the Tetanus and TC groups indicated a significantly higher rate of anti-tetanus Ab rise in the TC (*p* = 0.004). Therefore, the average rise in the antibody titer in people who received the PastoCovac vaccine after the Td vaccine was 36.8 units higher than in the people who received the Td vaccine. In other words, the average increase in the antibody titer in the TC group is nearly1.7 times more than the Tetanus group.

Furthermore, we merged the TC and CT results to compare the anti-tetanus Ab mean titer with the Td recipients. The results indicated a statistically significant difference (*p* = 0.006), in which the mean antibody titer rise in people who received the PastoCovac and Td vaccines was 26.8 units more than the Td group. That is to say, the ratio of the mean of the antibody titer rise in the CT + TC groups is almost 1.5 times of the Tetanus group.

Of the PCovac-1D group, one case was Ab-negative (Ab index: 4) before vaccination, in whom the PastoCovac injection led to a high antibody titer (Ab index: 74.4). Furthermore, there was a case who remained anti-tetanus Ab-negative from the PCovac-2D group. In the Control group who had a history of Td vaccination of more than 10 years, seven (33%) individuals were anti-tetanus-negative.

In total, a high titer of anti-tetanus was achieved in the PastoCovac recipients, which was comparable with the Td recipients. The best anti-tetanus induction was recorded in those that had a history of a Td vaccine and then received the PastoCovac injection. Furthermore, the individuals who received the PastoCovac vaccine as the primary series, and also as the booster one, reached a higher mean of the Ab than Td-vaccinated cases.

### 3.1. Antitetanus Persistency among PastoCovac Recipients

In order to explore the persistency of the anti-tetanus antibody induced by the PastoCovac vaccine, sera samples of PCovac-1D cases were collected before vaccination and 21, 90, and 180 days post-vaccination. According to Figure 2, a significant antibody titer was detectable from 21 to 180 days compared to the pre-vaccination time. Furthermore, the anti-tetanus Ab level on day 180 was still significantly high compared to day 21. Although the antibody titer on day 180 decreased to a lower level than on day 21, it is still estimated to be effective in the prevention from tetanus.

### 3.2. Assessment of Adverse Events (AEs)

According to the safety monitoring results, the only case who presented some sort of AE was among the Tetanus group, with a local pain at the injection site, headache, fever, and shivering after the Td vaccination. Nevertheless, there were no complaints after PastoCovac administration of any doses or the combination of the Td and PastoCovac vaccines (CT/TC).

## 4. Discussion

Tetanus vaccination has a primary-series schedule in childhood and the second decade of life (16 to 18 years old). Then, individuals should preferably receive the booster doses every 10 years. On the contrary, a large investigation showed that booster vaccines against tetanus or diphtheria are not necessary if adults have completed the profile of childhood vaccination series. They reviewed WHO data from 31 countries for 15 years, which included 11 billion person-years. No significant difference in infection rates was found between the countries which recommend adults to obtain booster shots and those that do not. Therefore, it was suggested that childhood vaccination alone could protect sufficiently against tetanus and diphtheria [13,14]. However, the question of having ongoing booster shots is really more complicated than only considering the frequency of a disease [13,15]. Although there has been a lack of change in tetanus incidence, other factors could strongly affect the number of cases, including wound management, the amount of bacteria in the environment, and hygiene measures. Even though the rate of infection with tetanus might be rare, people still experience serious or deadly effects. Owing to the fact that there is no specific treatment for tetanus, and no proof of having lifelong immunity with childhood vaccinations, the CDC still recommends booster vaccines every 10 years to help the immune system against the infection. The vaccination against tetanus through Td or Tdap protect > 95% of people for approximately 10 years according to CDC guidelines [13]. Not only do the booster vaccines against tetanus raise questions, coupled vaccines to the tetanus toxoid also bring concerns of immune response interference.

Recombinant subunit vaccine platforms provide us with some advantages, such as the safety, stability, the targeting the immune responses to the antigen of interest, and the ease of production [3,4]. PastoCovac has been the only COVID-19 subunit vaccine which is conjugated to the tetanus toxoid. There are some benefits of the vaccination with the coupled viral antigen to the tetanus toxoid, including the predominant IgG immune response caused by the affinity maturation and durable specific B-memory cells. The orientation of the RBD which is conjugated to the tetanus toxoid leads to a greater exposure to the receptor-binding motif and the increase in the level of neutralizing antibodies [16,17,18].

The impact of tetanus-conjugated vaccines on immune responses against tetanus as the carrier protein has remained unclear. Although the immunogenicity and effectiveness of conjugate vaccines regarding the target antigens have been widely investigated, their possible protection ability against tetanus has been overshadowed and less discussed.

Since the COVID-19 pandemic, different vaccine platforms have been designed to inhibit SARS-CoV-2 spread, among which PastoCovac has been the only tetanus-conjugated vaccine. This vaccine has been approved to administer in Iran as the primary and booster doses, and so there are immunized populations among whom some individuals received one, two, or three doses. This means the immune system has not only been induced against SARS-CoV-2 RBD, but also the tetanus toxoid. A similar status also applies to conjugate pneumococcal vaccines worldwide. In the present research, the PastoCovac vaccine was assessed to evaluate the anti-tetanus antibody induction and the comparability of the induced immune response with the Td vaccine. Furthermore, the safety of the Td and PastoCovac vaccines regarding the priority of injection and number of doses was considered.

It has been stated that he presence of previous immunity to a carrier protein could possibly suppress the immune induction to the conjugate, owing to carrier-induced epitopic suppression (CIES) [19,20]. One of the clinical studies regarding this issue was on the meningococcal serogroup C conjugate (MCC) vaccine, which contains TT. The results showed that protection against MenC in all three groups of children who received the DT/TT and then the MCC-TT, the co-administration of the DT/TT and MC-TT, and the DT/TT after the MC-TT was indicative, though the highest Ab rise was achieved in the first group. Bystander interference has been also proposed as a possible cause of the immune response reduction when two antigens are co-administered, which is defined as carrier-induced epitopic suppression [21]. The study conducted in Australia highlighted that immunization with the Tdap (tetanus, diphtheria, and pertussis) vaccine three to four weeks prior to the co-administration of the Pneumococcal conjugate vaccine (PCV13)/Meningococcal vaccine (MCV4) in adults significantly suppressed the subsequent antibody rise to some of PCV13’s serotypes. They concluded that carrier proteins may induce specific anti-carrier antibody responses which interfere with immune responses to the conjugate vaccine. They found it reasonable to assume that the Tdap antigens might negatively interact with some CRM197-conjugated PCV13 serotypes as a consequence of the similarity between the CRM197 carrier protein and the DT in the Tdap vaccine [22].

The antibody affinity, T-cell responses, and B-cell activation have been previously investigated, and the highly immunogenic potency of the RBD_TT/Alum formula in comparison with RBD alone was proven. In fact, multiple T-cell and B-cell epitopes of tetanus as the carrier induced better cellular immunity, as well as multimeric RBD-TT, which activated several B-cell receptors [18]. Further studies also confirmed the strong immune induction in immunized individuals with PastoCovac [23,24]. In other words, the administration of PastoCovac could simultaneously induce immune responses against SARS-CoV-2 and tetanus with no detectable interference. Furthermore, the administration of the Td vaccine prior to PastoCovac resulted in the highest anti-tetanus Ab rise in our study.

In an investigation in Korea, the comparison of the anti-tetanus antibody between two groups of individuals, including the Td vaccine + 13-valent pneumococcal conjugate vaccine (PCV13) and the Td vaccine alone, was performed. The results showed that Td injection generated significantly higher anti-tetanus antibodies than the Td + PCV13 co-administration [25]. There were two groups of individuals in the present study who received the Td vaccine before the PastoCovac vaccine, or after that. However, the two vaccines were not co-administered, and the results did not indicate any negative interaction between the two vaccines. Furthermore, the prior injection of the Td vaccine led to a greater anti-tetanus Ab induction compared to those who received PastoCovac first.

In the summaries of the product characteristics (SmPCs) and the leaflets of the conjugate vaccines, it is clearly stated that the use of these vaccines should not be considered instead of the Td vaccine [12]. This warning might be due to the lack of studies for the evaluation of specific immune response to the applied carrier protein. The in vivo experiment in mice or guinea pigs to show the capability of the carrier protein immune responses showed that the conjugate carriers TT and DT protected the immunized animals against a lethal challenge by the toxins. They also observed that a total amount of 10.5 µg of TT resulted in full protection after one injection of the conjugated vaccine against the tetanus lethal challenge. Interestingly, the reduced amount of TT, at 2.1 µg, also provided the same result. Moreover, the immunized animals with one dose of a conjugated vaccine containing 5 µg DT as the carrier resulted in 100% protection against the lethal challenge [12].

The PastoCovac vaccine contains 20 μg of TT as the carrier protein, which seems to present in a highly induced level, according to the serology assay, in the present data.

The vaccine data safety during the follow-up only indicated some AEs in a case who received the Td vaccine. According to the vaccine recommendations and guidelines of the Advisory Committee on Immunization Practices (ACIP), CDC, simultaneous or sequential vaccination is recommended when both are indicated. Our follow-up results clearly proved that the PastoCovac injection of any doses and the sequential administration of the PastoCovac and Td vaccines was safe, with no AEs.

Thus, considering the results of the present research is of a high value to clarify whether the immunization with a TT-conjugated vaccine could induce a booster response to tetanus. According to the present data, a positive immune response was verified in PastoCovac recipients, which is comparable with the Td recipients, and could lead to a reduced number of Td booster immunizations among individuals. The persistency of anti-tetanus Ab was sufficient; also, additional doses of PastoCovac led to a higher rate of this Ab. Nevertheless, this study was limited to a small population, and the persistency of the humoral immune response was subjected to only one group who completed the follow-up. This kind of study is required on larger populations to assess the protective potency of TT-conjugate vaccines and the possibility of reduced tetanus boosters in adults.

## 5. Conclusions

The current data indicate that the PastoCovac vaccine, composed with TT as the carrier protein, could induce humoral immune response against tetanus, which could persist for at least 6 months. Combination vaccine formulae containing TT and DT as carriers for conjugate vaccines could be developed instead of TT and/or DT alone. That is to say, a vaccine combination including different antigens could not only protect against the target antigens, such as hepatitis B and meningococcal serogroup, but also against the carrier proteins, like tetanus and diphtheria. Therefore, the free valence of TT and/or DT could be applied to add other antigen components which contribute protection against additional infectious diseases.

## Figures and Tables

**Figure 1 vaccines-12-00243-f001:**
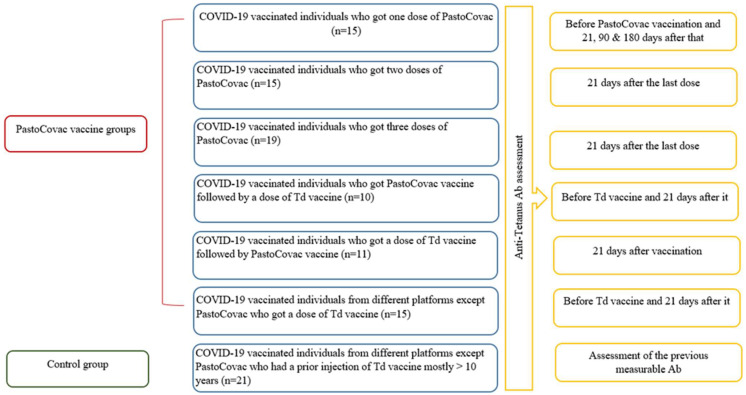
Study design. The recipients were divided into 6 groups according to the vaccine doses and the priority of the administered TD/PastoCovac vaccines. The other COVID-19 vaccine recipients with a history of a previous Td vaccine were considered as controls.

**Figure 2 vaccines-12-00243-f002:**
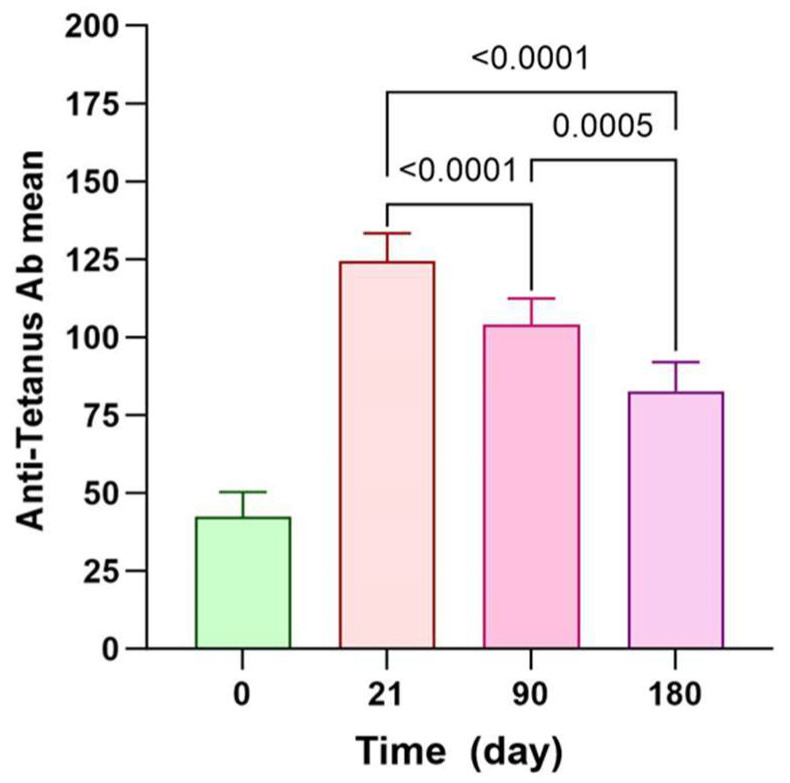
Antitetanus antibody persistency. The durability of the induced anti-tetanus Ab, which was evaluated by the comparison of the mean titer on day 90 and 180 with the day 21 level. Although the mean of the Ab titer declined over time, at 180 days post-injection, it was still at a high level.

**Table 1 vaccines-12-00243-t001:** Classification of the individuals to assess the PastoCovac immunogenicity against tetanus.

Study Groups	N	Mean ± SD	Median [IQR]	(Min–Max)
PCovac-1D	15	42.3 ± 31.2	26.8 [82.4–15.8]	(4–91.8)
PCovac-2D	15	74.3 ± 51	62.1 [109.6–33.5]	(7.6–167.5)
PCovac-3D	19	87 ± 28.9	97.1 [114.5–61.8]	(39.1–129.4)
CT	10	67.8 ± 26.4	62.1 [89.7–48.9]	(27.6–108.7)
TC	11	88.9 ± 37.6	99.2 [110.1–62.2]	(30.9–163.7)
Tetanus	15	52 ± 20.7	52.5 [72.1–32.2]	(14.2–79.3)
Control	21	29.9 ± 27.2	24.7 [38.3–9.6]	(4.2–107.8)

**Table 2 vaccines-12-00243-t002:** Comparison of the anti-tetanus Ab induction between the vaccinated groups.

Group	Group	*p* Value
Tetanus	CT	*p* = 0.109
Tetanus	TC	***p* = 0.004** *
Tetanus	CT + TC	***p* = 0.006**
CT	TC	*p* = 0.157
PCovac-1D	PCovac-2D	***p* = 0.061**
PCovac-1D	PCovac-3D	***p* = 0.003**
PCovac-1D	Tetanus	*p* = 0.326
PCovac-2D	Tetanus	*p* = 0.128
PCovac-3D	Tetanus	***p* < 0.001**

* Significant values are shown in bold.

## Data Availability

All the relevant data are included in the study.

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
