# Peer review of "Potential Protectivity of a Conjugated COVID-19 Vaccine against Tetanus"

_vaccines, 2024, doi:10.3390/vaccines12030243_

Round 1

Reviewer 1 Report

Comments and Suggestions for Authors

PastoCovac is a subunit protein vaccine against COVID-19 that has been used in Israel. This vaccine contains tetanus toxoid as a carrier conjugated to SARS-CoV-2-RBD. In this manuscript, the authors evaluated PastoCovac’s ability to induce immunogenicity against tetanus. The data indicated that such a vaccine triggers a strong and rather long-lasting immune response (>6 months). Overall, the results provide interesting insights into using an irrelevant vaccine containing a carrier as a vaccine against the disease the carrier is derived from.  

There are only some minor comments for the authors to address:

1. Lines 88 and 90, could the authors please provide the doses of PastoCovac used in groups 5 and 6 before or after the testers obtained their one Td vaccine dose? Once info is added, please also justify why the number of doses of PastoCovac was chosen for the groups.

2. It is suggested that the P values can be incorporated into Table 1 or perhaps, generate a figure of the ab titers against tetanus with the P values for the 7 tested groups.

Author Response

Reviewer 1

PastoCovac is a subunit protein vaccine against COVID-19 that has been used in Israel. This vaccine contains tetanus toxoid as a carrier conjugated to SARS-CoV-2-RBD. In this manuscript, the authors evaluated PastoCovac’s ability to induce immunogenicity against tetanus. The data indicated that such a vaccine triggers a strong and rather long-lasting immune response (>6 months). Overall, the results provide interesting insights into using an irrelevant vaccine containing a carrier as a vaccine against the disease the carrier is derived from.  

There are only some minor comments for the authors to address:

  1. Lines 88 and 90, could the authors please provide the doses of PastoCovac used in groups 5 and 6 before or after the testers obtained their one Td vaccine dose? Once info is added, please also justify why the number of doses of PastoCovac was chosen for the groups.

Thank you for the comment. In both groups, the PastoCovac was administered once. In CT group individuals got a dose of PastoCovac then a Td vaccine. In TC group, the cases got Td vaccine and then a shot of PastoCovac. We aimed at evaluation of association between the number of doses and anti-tetanus Ab induction. Therefore, the selected groups are divided according to the number of PastoCovac injection. The results showed that the increase in number of PastoCovac injection did not result in any adverse events in the individuals and led to a greater anti-tetanus Ab induction. The number of PastoCovac injection was added to the vaccinated groups in methods section accordingly.

  1. It is suggested that the P values can be incorporated into Table 1 or perhaps, generate a figure of the ab titers against tetanus with the P values for the 7 tested groups.

Table 2 has been added to show the p values between the groups.

Reviewer 2 Report

Comments and Suggestions for Authors

The manuscript entitled ''Potential Protectivity of a Conjugated COVID-19 Vaccine 2 against Tetanus'' presents an interesting piece of work depicting the omittance of anti-carrier antibodies as combination vaccine consisting of TT and DT. I have several issue related to the manuscript:

1. Introduction- well represented

2. Results are indicated clearly but the link between the different groups is missing.

3. The manuscript is based only on Anti-tetanus antibodies only which makes sense. Did the authors also tested for circulating IgG against COVID-19.

4. Did the authors set up an invitro assay to see the inhibition of COVID-19 antibodies on the production of anti-tetanus antibodies and vice-versa.

5. The authors can perform flow cytometry from PBMC to observe other markers indicating a strong anti-tetanus response.

6.  Anti-Tetanus persistency among PastoCovac recipients- The durability of the induced anti-tetanus Ab which 174 was evaluated by comparison the mean titer of day 90 and 180 with day 21 level. Although the mean 175 of Ab declined over time, its level 180 days post-injection is still protective.

How the authors know that this response will be protective or no. Do they have any of protection index based on anti-tetanus antibodies level. 

Author Response

The manuscript entitled ''Potential Protectivity of a Conjugated COVID-19 Vaccine 2 against Tetanus'' presents an interesting piece of work depicting the omittance of anti-carrier antibodies as combination vaccine consisting of TT and DT. I have several issue related to the manuscript:

  1. Introduction- well represented

Thank you for your time and great attention.

  1. Results are indicated clearly but the link between the different groups is missing.

The results have been refined according to the comment.

  1. The manuscript is based only on Anti-tetanus antibodies only which makes sense. Did the authors also tested for circulating IgG against COVID-19.

We do appreciate the comment. The aim of this study was determination of the PastoCovac ability in anti-tetanus Ab induction. The vaccine immunogenicity regarding SARS-CoV-2 antibodies have been massively investigated in our studies and also Cuban investigation which showed great protectivity against the virus [1-4].

  1. Did the authors set up an invitro assay to see the inhibition of COVID-19 antibodies on the production of anti-tetanus antibodies and vice-versa.

We do appreciate the comment. The preclinical assessment had been previously conducted and proved no inhibitory effect between the antibodies [5] (described in discussion section ref.19).

  1. The authors can perform flow cytometry from PBMC to observe other markers indicating a strong anti-tetanus response.

In this study, the level of anti-tetanus Ab induced by PastoCovac vaccine was compared to the Abs level of Td vaccine which showed the great potency of PastoCovac vaccine in anti-tetanus Ab induction. PBMCs were not collected in this investigation. Thank you for the comment. In further studies, we will possibly evaluate the anti-tetanus potency in larger population and will apply this method as well.

  1. Anti-Tetanus persistency among PastoCovac recipients- The durability of the induced anti-tetanus Ab which 174 was evaluated by comparison the mean titer of day 90 and 180 with day 21 level. Although the mean 175 of Ab declined over time, its level 180 days post-injection is still protective.

How the authors know that this response will be protective or no. Do they have any of protection index based on anti-tetanus antibodies level. 

This statement is according to the ELISA assay in which the positive level of anti-tetanus is >11 and also the comparison between PastoCovac result and the Td vaccine result. The positive antibody induction was replaced accordingly.

  1. Ramezani, A., et al., PastoCovac and PastoCovac Plus as protein subunit COVID-19 vaccines led to great humoral immune responses in BBIP-CorV immunized individuals. Scientific Reports, 2023. 13(1): p. 8065.
  2. Ashrafian, F., et al. A Comparative Study of Immunogenicity, Antibody Persistence, and Safety of Three Different COVID-19 Boosters between Individuals with Comorbidities and the Normal Population. Vaccines, 2023. 11, DOI: 10.3390/vaccines11081376.
  3. Eybpoosh, S.e.a., Immunogenicity and safety of heterologous boost immunization with PastoCovac Plus against COVID-19 in ChAdOx1-S or BBIBP-CorV primed individuals

PLOS Pathogens, 2023.

  1. Farahmand, B., et al., Evaluation of PastoCovac plus vaccine as a booster dose on vaccinated individuals with inactivated COVID-19 vaccine. Heliyon, 2023. 9(10): p. e20555.
  2. Valdes-Balbin, Y., et al., SARS-CoV-2 RBD-Tetanus Toxoid Conjugate Vaccine Induces a Strong Neutralizing Immunity in Preclinical Studies. ACS Chem Biol, 2021. 16(7): p. 1223-1233.